# Influenza Vaccination Assessment according to a Value-Based Health Care Approach

**DOI:** 10.3390/vaccines10101675

**Published:** 2022-10-08

**Authors:** Giovanna Elisa Calabrò, Floriana D’Ambrosio, Elettra Fallani, Walter Ricciardi

**Affiliations:** 1Section of Hygiene, University Department of Life Sciences and Public Health, Università Cattolica del Sacro Cuore, 00168 Rome, Italy; 2VIHTALI (Value in Health Technology and Academy for Leadership & Innovation), Spinoff of Università Cattolica del Sacro Cuore, 00168 Rome, Italy; 3Department of Life Sciences, University of Siena, 53100 Siena, Italy; 4Seqirus S.R.L., Via del Pozzo 3/A, San Martino, 53035 Monteriggioni, Italy

**Keywords:** influenza, flu, vaccination, vaccines, value, value-based health care, personal value, allocative value, technical value, societal value

## Abstract

Background: Seasonal influenza has a considerable public health impact, and vaccination is the key to preventing its consequences. Our aim was to describe how the value of influenza vaccination is addressed in the scientific literature considering a new value framework based on four pillars (personal, allocative, technical, and societal value). Methods: A systematic review was conducted by querying three databases. The analysis was performed on international studies focused on influenza vaccination value, and the four value pillars were taken into consideration for their description. Results: Overall, 40 studies were considered. Most of them focused on influenza vaccination in the general population (27.5%), emphasizing its value for all age groups. Most studies addressed technical value (70.4%), especially in terms of economic models and cost drivers to be considered for the economic evaluations of influenza vaccines, and societal value (63%), whereas few dealt with personal (37%) and allocative values (22.2%). Conclusions: The whole value of influenza vaccination is still not completely recognized. Knowledge and communication of the whole value of influenza vaccination is essential to guide value-based health policies. To achieve this goal, it is necessary to implement initiatives that involve all relevant stakeholders.

## 1. Introduction

Every year, seasonal influenza (or flu) affects approximately 1 billion people of all age groups worldwide, and it has a considerable public health impact that results in an increased economic burden on both the health system and society [1]. In particular, seasonal influenza epidemics cause approximately 3–5 million severe cases annually, especially among vulnerable groups, such as elderly individuals, younger children (<5 years of age), pregnant women, and individuals with chronic diseases, and 290,000 to 650,000 deaths globally [2]. Vaccination is the key to preventing influenza and its consequences and to reducing its clinical and economic burden on health care systems and society [3].

Health care systems are constantly in search of effective primary prevention strategies, and in recent years, several new flu vaccines have been produced to deliver health benefits and protect communities [4]. However, vaccine efficacy, effectiveness, and safety assessment include only the minimum information needed for regulatory approval, rather than the full public health value of vaccines, and vaccine efficacy and effectiveness are usually focused on the direct protection of the vaccinated individual [5]. The full benefits of vaccination go beyond direct prevention of etiologically confirmed diseases; vaccination prevents community outcomes through indirect protection, contributes to health system sustainability through savings generated in terms of reductions in hospitalizations, direct medical costs, and any short- and long-term complications related to infectious diseases, and also supports health equity and national economies by reducing the loss of productivity due to absence from work and maintaining people’s health [5,6]. Thus, economic evaluations should consider these broad benefits [7].

To date, even the flu vaccination has not been evaluated to fully account for all its benefits. However, influenza vaccination has a substantial whole value, as it has personal value because it guarantees protection to individuals [8]; it has considerable societal value because the vaccination of the individual or of some groups at risk, such as children and health care workers (HCWs), protects the most vulnerable people, such as elderly individuals [9,10,11]; influenza vaccination is cost-effective [12]; it can also reduce indirect costs related to the loss of productivity of workers and caregivers [3]; and it has an important fiscal impact, as guaranteeing the health of workers increases the tax revenues of the state and the propensity to consume of workers and their families [13].

Therefore, new data and evidence-based instruments are needed to support the decision-making process on vaccines, such as the value-based health care (VBHC) approach, the health technology assessment (HTA) and new health economic models, in order to assess the whole value of vaccinations [6].

Recently, the Expert Panel on Effective Ways of Investing in Health (EXPH) of the European Commission (EC) proposed a VBHC approach based on four value pillars: personal value, allocative value, technical value, and societal value [14]. These pillars are the guiding principles of solidarity-based health care systems. In 2022, we published a study [15] that aimed to identify and systematically describe how the value of vaccines and vaccination was addressed in the scientific literature considering the four EXPH value pillars framework. In this study, we documented how vaccine evaluation is still limited to a few pillars and does not yet consider the whole value of vaccination.

Similarly, the aim of this new study is to investigate, considering the four EXPH value pillars framework, how the international scientific community addresses and evaluates the values of influenza vaccination. Describing the scientific evidence on the values of influenza vaccination will be useful in promoting new health policies that consider the whole value of vaccination.

## 2. Materials and Methods

A systematic literature review was performed considering the methodological approach used in our recent study and applying it to influenza vaccination [15]. An in-depth analysis was performed on studies that addressed value as a key element, and the four value pillars (personal, allocative, technical, and societal) proposed by the EXPH of the EC [14] were taken into consideration for their description.

The systematic review was conducted according to the Preferred Reporting Items for Systematic Reviews (PRISMA) guidelines [16].

### 2.1. Search Strategy

The literature search was performed by consulting three databases, namely, PubMed, Web of Science (WoS), and the University of York’s Center for Reviews and Dissemination (CRD) database. The search strings were launched on 5 July 2022. The systematic review was performed from 24 December 2010, onward, as Michael Porter’s first paper on value in health care was published on 23 December 2010 [17].

The following search string was used in PubMed:

((“value”[All Fields] OR “values”[All Fields]) AND (“vaccin”[Supplementary Concept] OR “vaccin”[All Fields] OR “vaccination”[MeSH Terms] OR “vaccination”[All Fields] OR “vaccinations”[All Fields] OR “vaccines”[MeSH Terms] OR “vaccines”[All Fields] OR “vaccine”[All Fields]) AND (“influenza, human”[MeSH Terms] OR (“influenza”[All Fields] AND “human”[All Fields]) OR “human influenza”[All Fields] OR “influenza”[All Fields] OR “influenzas”[All Fields] OR “influenzae”[All Fields] OR “flu”[All Fields])).

This spelling was then adapted to the WoS and CRD databases. The following filters were applied: human studies and English language. The article records were entered into an Excel worksheet and screened according to the inclusion/exclusion criteria. A check for duplicates was performed; the selection was made first by reading titles and abstracts and then the full texts.

### 2.2. Inclusion/Exclusion Criteria

Studies on influenza vaccination and flu vaccines that mentioned the term “value” in any part of the text and were conducted internationally were considered potentially eligible. We included original articles, literature reviews, systematic reviews, and expert opinions exclusively in the English language and published as of 24 December 2010. Commentaries, editorials, conference presentations, and references that were not provided with full text, as well as studies conducted on animals or in vitro, were excluded.

### 2.3. Selection Process and Data Extraction

Two researchers (F.D. and E.F.) independently screened titles and abstracts first and full texts afterward. Any disagreement was resolved by discussion or by the involvement of a senior researcher (G.E.C.).

Furthermore, the included studies were subjected to the snowballing process using bibliographic references to identify additional articles that met the inclusion criteria of our review.

From the articles definitively included in the literature review, the following information was extracted: first author’s name, publication year, study perspective (European, non-European, or global perspective), study aim, type of study, and target population of the influenza vaccination. In addition, for all included studies, information was collected and systematized on the main dimensions of the four value pillars considered (personal, allocative, technical, and societal) and on other aspects of the influenza vaccination value possibly addressed in the studies. Finally, for each article, the main reflections/actions that emerged on flu vaccination were summarized.

## 3. Results

The overall research in the three databases yielded a total of 1851 articles. After duplicates removal, 1450 articles were screened based on the title and abstract. In total, 59 full-text articles were selected. Following the inclusion and exclusion criteria, the screening resulted in the final inclusion of 40 articles. No new studies were included after the snowballing process. Details about the study selection process are shown in Figure 1.

Of the 40 studies included in our systematic review [3,4,5,8,10,11,12,13,18,19,20,21,22,23,24,25,26,27,28,29,30,31,32,33,34,35,36,37,38,39,40,41,42,43,44,45,46,47,48,49], seven (17.5%) had a non-European perspective and were conducted in Canada (*n* = 1) [18], the USA (*n* = 3) [19,21,42], China (*n* = 1) [36], Iran (*n* = 1) [49], and low-income countries (LICs) (*n* = 1) [32]; ten (25%) had a global perspective [5,10,12,20,22,24,25,37,40,48]; 22 (55%) had a European perspective [3,4,8,11,13,23,26,27,28,30,31,33,34,35,38,39,41,43,44,45,46,47]; and one study (2.5%) had both a European and USA perspective [29]. Among the European studies, eight (36.4%) were conducted specifically in Italy [3,4,13,31,38,39,43,47]. In addition, of the 40 articles included in our work, 17 (42.5%) were systematic reviews or literature reviews [3,5,10,11,12,22,24,25,28,29,30,31,32,37,41,47,48], ten (25%) were economic evaluations [8,13,19,21,23,26,27,34,44,45], seven (17.5%) were cross-sectional studies [18,35,36,38,42,43,49], two studies (5%) were HTA reports [4,39], one study (2.5%) was a nonlinear regression model [33], and three studies (7.5%) were expert opinions [20,40,46]. The main features of the studies are shown in Table 1.

Regarding the influenza vaccination target population investigated in the studies included in our systematic review, 11 studies (27.5%) focused on vaccination in the general population [5,12,18,20,25,37,39,40,44,46,48], five (12.5%) on vaccination in the pediatric population [3,19,29,33,45], three (7.5%) on vaccination in the population aged 6 months to over 85 [21,27,28], seven (17.5%) on vaccination in elderly individuals (over 65) [4,11,24,30,31,41,47], two (5%) specifically on adults over 18 [38] and over 50 [26], one study (2.5%) specifically on influenza vaccination in elderly individuals and at risk groups [22], one (2.5%) on the target population for which the WHO recommends flu vaccination [8], one (2.5%) on customs officers [23], four (10%) on HCWs [10,34,35,36], one (2.5%) on workers in general [13], two (5%) on health care students [42,43], one (2.5%) on pregnant women [32], and another study (2.5%) on influenza vaccination in the cancer population [49].

Table 2 reports the main results on influenza vaccination values and the main reflections/actions that emerged from studies included in our systematic review.

Regarding the focus on the influenza vaccination value, 67.5% (*n* = 27) of the included studies investigated aspects relating to the four pillars of value (personal, allocative, technical, and societal). Specifically, in this group of studies, 37% of them (*n* = 10) addressed the issue of personal value, with particular interest in clinical outcomes and citizen involvement/empowerment [3,4,5,8,20,30,32,35,46,49]; 22.2% (*n* = 6) dealt with the issue of allocative value in terms of accessibility and equity of access to influenza vaccination [4,5,18,20,32,46]; 70.4% (*n* = 19) investigated technical value, especially in terms of economic models and cost drivers to be considered for the economic evaluations of flu vaccines [3,4,5,8,12,13,19,21,22,23,27,28,29,31,34,44,45,46,48]; and 63% (*n* = 17) of the studies dealt with the issue of societal value, with a particular focus on population wellbeing and the indirect protection of the community [3,4,5,8,10,12,18,20,30,31,32,34,35,36,45,46,48]. Among these, only one study addressed the issue of societal value linked to the dimension of shared decision-making [18]. Of the 27 articles, three (11.1%) addressed all the value pillars that applied to influenza vaccination [4,5,46].

Of the 40 studies included in the systematic review, 37.5% (*n* = 15) also addressed other general features of the influenza vaccination value [11,24,25,26,33,36,37,38,39,40,41,42,43,47,49], with particular reference to the following aspects: the cultural value and social benefits of flu vaccination both from the citizens’ perspective [24] and from that of health care professionals [36]; the value of influenza vaccination in specific target populations such as elderly individuals [11,26,41], adults [47], risk groups, such as patients with cardiovascular [41] and cancer [49] diseases, and health care students [43]; the importance of the whole value of flu vaccination [25] and the need for an appropriate methodology to assess this value and support an evidence-based decision-making process [37], also thanks to evidence-based tools such as HTA [39]; the value of surveillance systems for influenza and the need to consider new innovative surveillance methods such as social media [33]; the value of educational interventions on influenza vaccination for nursing students [42]; and the value of citizens’ literacy [38].

Eventually, Ruscio et al. [40] emphasized the need for globally coordinated planning to implement influenza vaccination, led by an alliance of international stakeholders, including representatives of governmental and nongovernmental organizations, representatives of civil society, industry, international organizations, and experts in health security and flu vaccination.

## 4. Discussion

Even if the benefits of flu vaccination are recognized worldwide, in several countries, including Italy, there is still inadequate vaccination coverage [15]. Furthermore, as demonstrated by our systematic review, the whole value of influenza vaccination is still not completely recognized.

Our review summarizes the currently available evidence on the value of influenza vaccination, considering the value perspective proposed by the EXPH of the EC. Our results showed that the personal, technical, allocative, and societal values of influenza vaccination were addressed by a limited number of studies in the last 12 years. Only three studies addressed the whole value of flu vaccination [4,5,46].

The issue of flu vaccination value was addressed by the scientific community, especially at the European level, with an important contribution from Italian researchers.

Regarding the target population of influenza vaccination investigated in the articles included in our systematic review, most of the studies focused on vaccination in the general population and on vaccination in elderly individuals, emphasizing the flu vaccination value in this age group. In fact, aging is associated with an increased risk of infectious diseases. The latter, in turn, due to the immunosenescence phenomenon and frequent comorbidities, is correlated with a greater risk of complications, hospitalizations, disability, and mortality [11]. Therefore, flu vaccination results in savings in health care and societal costs and represents a valuable intervention to prevent the disease in this population. However, despite the availability of vaccines, vaccination coverage rates are still suboptimal among elderly individuals, and they must be improved to achieve the full benefits of vaccination worldwide [11]. Furthermore, protection for elderly individuals may be improved by offering flu vaccination to younger individuals, caregivers, and others who are in contact with elderly individuals [30].

Of note, most of the studies focused on influenza vaccination in the general population, emphasizing the importance and value of this vaccination for all age groups [48].

If we consider the in-depth study on the influenza vaccination value, the results of our review show that most of the studies investigated aspects related to at least one of the four value pillars (personal, allocative, technical, and societal), and less than half dealt with the issue of the influenza vaccination value in general.

Specifically, the most investigated value pillar was the technical pillar, and this result is in line with what was also documented in our previous study on the value of vaccination [15]. The studies that investigated this pillar were mainly economic models of flu vaccines, developed from both the perspective of the health system and from the societal perspective. Several studies stressed the importance of including the societal perspective in economic evaluations of influenza vaccines, precisely in light of the societal value of this vaccination. Furthermore, de Waure et al. [22] pointed out that influenza vaccination in elderly individuals and in high-risk groups is a cost-effective intervention from a pharmacoeconomic point of view; however, a standardization of the methodology applied in the economic evaluations of influenza vaccines is needed to ensure comparability and transferability of the results. In contrast, Ting et al. [12] reported that, from a societal perspective, influenza vaccination is cost-effective for children, pregnant and postpartum women, high-risk groups, and in some cases, healthy working-age adults. In addition, according to the authors, for the economic evaluation of influenza vaccines, it is also necessary to consider the benefit of herd immunity linked to the increase in vaccination coverage.

Interesting data also emerged from the study by Wilder-Smith et al. [5], in which, according to the authors, the economic impact of influenza vaccination should incorporate the health and no health benefits of vaccination, both in the vaccinated and in the unvaccinated population, thus also allowing for estimation of the societal value of vaccination. The full benefits of vaccination, according to the authors, go beyond the direct prevention of etiologically confirmed diseases and often extend throughout a vaccinated person’s lifetime, prevent community outcomes, stabilize health systems, and promote health equity, and they also lead to benefits for local and national economies. The authors also point out that for the economic evaluation of influenza vaccines, as well as other vaccines, dynamic economic models should have priority over static models, as the latter usually underestimate the effectiveness and cost-effectiveness of immunization programs, as they underestimate their indirect effects [5].

Moreover, health care decision-makers and policy-makers should be aware of the limitations of traditional economic evaluations for assessing vaccine value [46]. Future economic assessments should pay more attention to the effect of vaccination on complication prevention, the generation of health benefits for HCWs, and advantages for the community beyond individual protection; in addition, economic assessments must be conducted considering the societal perspective as well as that of the health system to underline and document the whole value of vaccines [46].

However, the consequences of a disease, the number of clinical events, and productivity losses disproportionally affect different subgroups of the population based on age or socioeconomic status: the burden of these events falls more heavily on disadvantaged groups [46]. Economic assessments should consider how the health costs and outcomes associated with vaccination contribute to broader goals of societal equity and guaranteeing health gains for the whole population [46]. Therefore, the development of new economic models capable of capturing not only the mere cost-benefit of flu vaccination but also the broader value of vaccination as an investment in health is needed [25]. In this perspective, the elaboration of standardized guidelines for the economic evaluation of the whole value of vaccinations is required to simultaneously grant a comprehensive assessment of vaccine worth and an objective estimation of the quality, validity, and reliability of the evaluations.

Furthermore, it was also pointed out that decision-makers should adopt a full societal perspective to assess the economic value of vaccines. Traditional methods to estimate the cost of illness from a societal perspective can also be improved by considering the fiscal impact, which explains the decrease in fiscal revenues due to disease [13]. The potential reduction in the fiscal impact associated with immunization strategies should be included in the assessment of the effects of new flu vaccines, adding this new dimension to their valorization [13]. Ruggeri et al. [13] evaluated the fiscal impact of a vaccination program for flu in Italy. Their model estimated that, based on 2.1 million infected people per year, the fiscal impact and social costs associated with influenza were EUR 160 million and EUR 840 million, respectively. A vaccination strategy resulting in a reduction in the number of infected people by 200,000 would lead to a decrease in productivity loss of EUR 111 million and an increase in tax revenue of nearly EUR 18 million annually [13].

Another interesting finding that emerged from our systematic review is that over 60% of the included studies addressed the issue of the societal value of flu vaccination. This value pillar was investigated, above all, as wellbeing for the population and as indirect protection of the community. Only one study addressed the issue of societal value linked to the dimension of shared decision-making [18]. In particular, the authors stressed the importance of formulating immunization plans based on a fair and shared process. In fact, conducting prepandemic research is essential to engage the public, educate them, and solicit citizen feedback [18].

Compared to our previous work [15], which showed that the societal value of vaccinations was still poorly investigated by the scientific community, in this current review, this value was one of the main values being recognized and associated with influenza vaccination.

Recognizing the broader consequences of influenza and also considering the consequences for society are essential for determining the full burden of influenza in different subpopulations and for assessing the whole value of interventions for prevention, including vaccination [48].

From the perspective of the societal value of influenza vaccination, particular attention has been given to the protection of vulnerable people. For example, McElhaney et al. [30] highlighted the importance of vaccinating the youngest and healthiest people before they become fragile, as well as protecting the most vulnerable people.

The implementation of flu vaccination for HCWs and people who are in contact with elderly individuals has also been indicated as a fundamental action to improve the protection of the community. Indeed, health care professionals and public health decision-makers should more deeply understand the value of flu vaccination and consider it an important preventive tool for promoting healthy aging [30]. Influenza poses a serious threat to public health, especially for vulnerable populations such as elderly individuals. Therefore, it is necessary to emphasize the societal and economic values of this vaccination [31]. Communication and awareness of the value of VPD prevention in the general population is an important starting point, and all health and social workers can play a key role in this [31]. Similarly, according to Boey et al. [35], it is necessary to properly plan campaigns against seasonal flu in which vaccination education, the communication of the benefits of vaccines and the vaccination value, and the implementation of an easily accessible vaccination are promoted. It is also important to focus not only on value for patients but also on personal benefits for HCWs [35]. The societal value of influenza vaccination is mainly related to the protection of others by vaccinated HCWs, even if, unfortunately, adequate vaccination coverage is not reported in this category [10].

Furthermore, the societal value of flu vaccination correlates perfectly with the vaccination of pregnant women, as maternal immunization prevents the disease in two high-risk groups: mothers and their babies during the first months of life [32]. Strengthening maternal immunization is essential, especially in developing countries, as this preventive intervention could improve health systems for prenatal care, favor the construction of a platform for the production of other vaccines to be used during pregnancy, and strengthen health systems in response to future pandemics by increasing the distribution of flu vaccines [32].

Another key value of influenza vaccination is personal value, which was addressed in just under 40% of the studies included in our systematic review. The personal value was emphasized, above all, in relation to the individual benefits on clinical outcomes related to vaccination and citizen involvement/empowerment.

Ultimately, only a few studies addressed the issue of allocative value in terms of accessibility and equity of access to influenza vaccination. This result is in line with what was also documented in our previous work on the vaccination value [15] and draws attention to the need to develop further research on this value dimension that is relevant to both health systems and citizens.

It is interesting to note that in the evaluation of the general value of influenza vaccination, not directly investigated according to the four pillars of values proposed by the EXPH, issues such as the cultural value and societal benefits of vaccination were addressed both from the perspectives of citizens [24] and from that of health professionals [36], as well as the value of vaccination in “special” populations such as elderly individuals [11,26,41], adults [47], patients with cardiovascular diseases [47] and those with cancer [49], and health care students [43].

Furthermore, in the studies that dealt more generally with the issue of the influenza vaccination value, particular attention has been given to the importance of the whole value of this vaccination [25] and the need for an appropriate methodology to evaluate its whole value and support process evidence-based decision-making [37], thanks to rigorous tools such as HTA [39]. Improved interpretation of IVE [37] and the use of appropriate immunogenicity measures of influenza vaccines [50] would also improve the broader assessment of the influenza vaccination value. Further information on the various modifiers of the immune response induced by influenza vaccines could be useful to better understand the broader value of vaccination, especially in particular risk groups [51]. Furthermore, the principles of personalized medicine have already been applied to the vaccinology field to better understand interindividual variations in vaccine-induced immune responses and vaccine-related adverse events. This knowledge could substantially improve the understanding of the onset of infections in people at risk and help determine the type or dose of vaccine needed [52].

Moreover, the value of influenza surveillance and the need to apply innovative surveillance methods through social media were also emphasized [33]. Another aspect of the value proposed by the scientific literature is that of the value of educational interventions in the field of influenza vaccination for nursing students [42] and of literacy for citizens [38]. Ultimately, it was emphasized that globally coordinated planning to implement flu vaccination around the world led by an alliance of international stakeholders (i.e., representatives of governmental and nongovernmental organizations, citizens, industry, international organizations, and experts in health security and influenza) is needed [40].

From what emerged from our systematic review, it is evident that knowledge and communication of the whole value of influenza vaccination is fundamental and necessary to guide health policies in the field of evidence-based and value-based immunization. However, there are several barriers to overcome in order to increase influenza vaccination coverage internationally, including the concerns of individuals about the safety of vaccines and their adverse events, lack of confidence in vaccinations, exposure to false myths that undermine trust in vaccines, the inability of some health care professionals to counter these myths and provide evidence and adequate informed advice, and structural and organizational barriers to fair access to vaccination [53]. Therefore, the actions to be implemented to increase vaccination coverage should focus on communication strategies concerning the benefits of vaccination and on greater dialogue—with a participatory approach—with the most hesitant groups on vaccines and vaccinations [54]. Following proper education, further support must be provided by an active invitation to vaccinate, conveyed through message or phone call, reminding the population to be vaccinated periodically [55,56]. Regardless, a greater involvement of general practitioners and pediatricians is needed to improve vaccination coverage rates in all age groups. However, better access to vaccination could be achieved with the involvement of other adequately trained health care professionals (e.g., nurses, pharmacists, etc.), thereby ensuring greater equity of access [53]. Another priority action is to strengthen monitoring and surveillance systems at the international, national, and local levels to ensure updated data to guide health policy and planning and thus implement vaccination coverage and provide a more reliable trend of the burden of disease [54]. Furthermore, understanding the whole value of vaccination and the effective translation of this knowledge for all stakeholders is essential to strengthen health policies and immunization strategies globally, as well as to combat vaccine misinformation and vaccination hesitancy [6,15].

Several limitations should be considered in our study. Only articles published in English until 5 July 2022, were included, which might have led to the failure to identify all the available evidence on the value of influenza vaccination. Moreover, selection bias could not be completely ruled out even though the screening process was performed rigorously and according to the PRISMA statement. It cannot be excluded that some dimensions of the value may have been missed, as the synthesis of evidence was based on the content of the document published by the EXPH in 2019. Furthermore, a quality assessment of the included studies was not performed, and we could not assess the methodological correctness of the included articles. However, in our opinion, this does not prejudice our work, as we wanted to provide an overview of the evidence on the values of influenza vaccination without addressing the robustness of the methods used to do so.

Moreover, it is important to underline that the estimation of the whole value of the influenza vaccination is particularly complex and, therefore, the scarcity of evidence recovered in the literature, especially for some value pillars, could be linked to this difficulty of evaluation. Our paper describes the current evidence on the assessment of personal, allocative, technical, and societal value of influenza vaccination and stresses the need to expand these aspects as well in the research field. As presented, the assessment of the whole value of flu vaccination needs to consider not just the direct impact on health and health care but also the wider impact on economic growth and societies. These wider impacts, although difficult to measure and still under-investigated, should be taken into consideration to better depict the whole value of flu vaccines and vaccination and to counteract vaccines hesitancy and misuse.

Finally, the heterogeneity of the evidence limits the possibility of delving into the data coming from the studies and of releasing definite results. However, in our opinion, our study could help advance the evaluation of the whole value of influenza vaccination to support a value-based decision-making and to promote innovative immunization strategies centered on the broader value of influenza vaccination.

## 5. Conclusions

In the context of vaccination prevention, particular attention must be paid to influenza vaccination, as influenza represents a public health problem with a considerable impact from an epidemiological, clinical, economic, and societal point of view. Infectious diseases do not recognize geographical and/or political borders, but especially those preventable by vaccines such as influenza require a global and not a local approach for their prevention and control. These strategies necessarily require the removal of ideological and political barriers but also of economic and cultural obstacles in favor of a global approach in defense of population health.

Based on what emerged from our review, there is a clear need to consider a value-based strategy of immunization against influenza, with the aim of concretely putting citizens and patients at the “center”. To do this, it is necessary to know and disseminate scientific evidence on the whole value of influenza vaccination, as well as to promote and implement immunization strategies that consider the broader and, therefore, the personal, technical, allocative, and societal values of vaccination.

Furthermore, addressing the recognition of the whole value of influenza vaccination could contribute to greater acceptance of the flu vaccines by the population and, therefore, increase vaccination coverage in all age groups.

Therefore, health care professionals, the scientific community, institutions, and decision-makers must commit themselves, each with their own skills, to promote the correct use of flu vaccines and to safeguard the undisputed whole value of flu vaccination and the heritage of all citizens, regardless of social status or place of residence.

## Figures and Tables

**Figure 1 vaccines-10-01675-f001:**
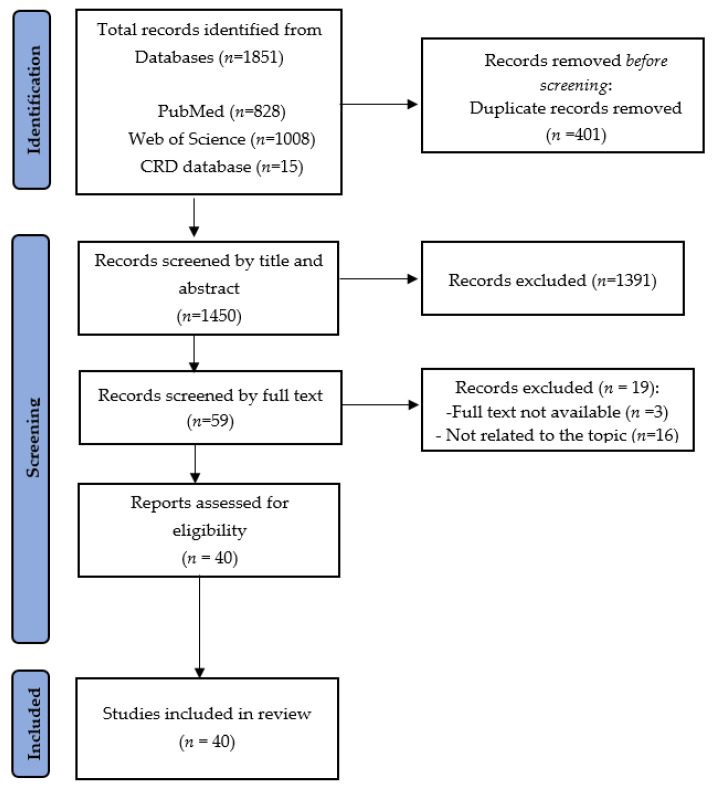
PRISMA statement flow diagram.

**Table 1 vaccines-10-01675-t001:** Main characteristics of the included studies.

First Author, Year [Ref]	European Perspective, Non-European Perspective or Global Perspective (Country)	Study Aim	Type of Study	Target Population of Influenza Vaccination
Bailey, T.M., 2011 [18]	Non-European perspective (Canada)	To investigate the views of university students, support staff, and academic staff on resource allocation during an influenza pandemic	Cross-sectional web-based survey	General population
Lee, B.L., 2011 [19]	Non-European perspective (USA)	To estimate the economic value of a “universal” influenza vaccine compared to the standard annual influenza vaccine, starting vaccination in the pediatric population	Economic evaluations from a societal perspective	Pediatric population (ages 2–18 years)
Luyten, J., 2011 [20]	Global perspective	To explore the following three policy questions: (i) Ethically, which policy measures should be addressed when vaccination coverage is insufficient in a population? (ii) Is it ethical to target vaccination programs at certain risk groups? (iii) What is the ethical significance of adverse herd immunity effects?	Expert opinion	General population
Lee, B.Y., 2012 [21]	Non-European perspective (USA)	To determine the economic value of a QIV compared to the TIV for ten influenza seasons (1999–2009) in the USA	Economic evaluation from a societal and a third-party payer perspective	From six-month-old children to over 85-year-olds
de Waure, C. 2012 [22]	Global perspective	To assess the economic value of influenza vaccination among elderly and high-risk groups	Systematic review	Elderly and high-risk groups
Mamma, M., 2013 [23]	European perspective (Greece)	To estimate the economic impact of the influenza vaccination program among customs officers in Greece during the 2009/2010 period	Decision analytical computational simulation model	Customs officers
Nagata, J.M., 2013 [24]	Global perspective	To assess the social determinants of health preventing adults ≥65 years of age from accessing and accepting seasonal influenza vaccination	Systematic review	Elderly individuals
Preaud, E., 2014 [8]	European perspective (27 European countries: Austria, Belgium, Bulgaria, Cyprus, Czech Republic, Denmark, Estonia, France, Finland, Germany, Greece, Hungary, Ireland, Italy, Latvia, Lithuania, Luxembourg, Malta, the Netherlands, Poland, Portugal, Romania, Slovakia, Slovenia, Spain, Sweden, UK)	To generate a model to assess the public health benefits and economic importance of influenza vaccination in the 5 WHO-recommended vaccination target groups in 27 countries of the EU	Economic evaluation from a societal and a third-party payer perspective	WHO-recommended vaccination target groups (children 6–23 months of age; individuals with underlying chronic health conditions; pregnant women, health care workers; elderly individuals over 65 years of age)
Rappuoli, R., 2014 [25]	Global perspective	To review the vaccine history, including influenza vaccines, the progress already achieved by vaccine technologies, and the potential that vaccines may have to prevent and cure the diseases of modern society across all age groups and all countries	Literature review	General population
Kristensen, M., 2016 [26]	European perspective (the Netherlands)	To estimate the disease burden for influenza, pertussis, PD, and HZ among adults aged 50 years or over in the Netherlands	Disease burden model using DALY measures	Adults aged 50 years or over
Uhart, M., 2016 [27]	European perspective (France, Germany, Italy, Spain and UK)	To estimate the public health and economic impact of seasonal influenza vaccination with QIVs compared to TIVs in Europe	Economic evaluation from a societal and a third-party payer perspective	From six-month-old children to over 65-year-olds
Barbieri, M., 2016 [28]	European perspective (Belgium, France, Germany, Italy, Spain, Sweden, the Netherlands, UK)	To review published cost-utility analyses of influenza vaccination strategies in eight European countries and to assess whether there are differences in cost-effectiveness terms among countries	Systematic review	From six-month-old children to over 65-year-olds
Gibson, E., 2016 [29]	European and non-European perspectives (Europe and USA)	To compare the economic value of pediatric immunization programs for influenza to those for RV, MD, PD, HPV, Hep B, and varicella reported in recent (2000 onward) cost-effectiveness studies	Systematic review	Pediatric population
McElhaney, J.E., 2016 [30]	European perspective (France, Belgium)	To examine the role of vaccination in elderly individuals	Literature review	Elderly individuals
Poscia, A., 2016 [31]	European perspective (Italy)	To sum up the key elements of influenza vaccination sustainability in Italy and to make suggestions for improving the organizational structure of vaccination initiatives	Literature review	Elderly individuals
Wilder-Smith, A., 2017 [5]	Global perspective	To propose a broader scope of methods, measures, and outcomes to evaluate the effectiveness and public health impact of vaccines to be considered for evidence-informed policy-making at both the pre- and post-licensure level	Literature review	General population
Lorenc, T., 2017 [10]	Global perspective	To synthesize evidence on HCWs’ perceptions and experiences of influenza vaccination	Systematic review	HCWs
Ting, E.E.K., 2017 [12]	Global perspective	To review the cost-effectiveness of influenza immunization programs to inform policy	Systematic review	General population
Ortiz, J.R., 2017 [32]	Non-European perspective (LICs)	To review the strategy of maternal influenza immunization for potential investment in LICs	Literature review	Pregnant women
Wagner, M., 2017 [33]	European perspective (England)	To assess the population impact of the programs launched in England during the 2013/2014 and 2014/2015 flu seasons	Nonlinear regression model that was trained to infer ILI rates from Twitter posts for the influenza season of 2014/2015	Pediatric population
Esposito, S., 2018 [11]	European perspective (Italy, France, Spain, Germany, UK)	To estimate the burden of VPDs (influenza, PD, HZ) among elderly individuals in Europe and summarize the potential public health benefits of vaccination strategies for these individuals	Literature review	Elderly individuals
Meijboom, M.J., 2018 [34]	European perspective (the Netherlands)	To assess the health economic value of implementing an influenza immunization program among HCWs	Cost-benefit model from a societal perspective	HCWs
Boey, L., 2018 [35]	European perspective (Belgium)	To determine demographic, behavioral, and organizational factors that are associated with vaccination uptake among HCWs in both hospitals and nursing homes	Cross-sectional survey	HCWs
Li, K.K., 2019 [36]	Non-European perspective (China)	To examine whether and in what way two individual-level cultural dimensions, collectivism, and power distance would influence vaccination via social benefits (i.e., self-and-clan protection and community protection) and social influence (i.e., authority advice and family-and-peer advice), respectively, among nurses	Cross-sectional online survey	HCWs (nurses)
Hollingsworth, R., 2020 [37]	Global perspective	To define an evaluative framework, which would include specific elements (study outcomes and setting, study design, confounding factors) to ensure the limitations of estimates of IVE, as an indicator of public health benefit that is fully appreciated and effectively communicated	Systematic review	General population
Domnich, A., 2020 [38]	European perspective (Italy)	To assess and describe the beliefs, attitudes, and practices of a representative sample of Italian adults regarding influenza vaccination	Cross-sectional survey	Adults (≥18 years old)
Calabrò GE, 2020 [39]	European perspective (Italy)	To describe how HTA has been incorporated as an evidence-based tool to support the definition of Italian vaccination strategies	HTA	General population
Ruscio, B., 2020 [40]	Global perspective	To determine the value, roles, and structure of an alliance of stakeholders that supports and promotes using sustainable seasonal influenza vaccination programs as a tool to create reliable and scalable pandemic vaccine programs globally	Expert opinion	General population
Aidoud, A., 2020 [41]	European perspective (France)	To review the literature data on the cellular mechanisms that link influenza vaccination to the prevention of atherosclerotic complications	Literature review	Elderly individuals
Valentino, S., 2020 [42]	Non-European perspective (USA)	To evaluate if an evidence-based influenza and vaccine education intervention will affect nursing students’ intent to vaccinate for influenza by increasing knowledge of the influenza vaccine	Cross-sectional computer survey	Nursing students
Cella, P., 2020 [43]	European perspective (Italy)	To explore health care students’ vaccination behavior and beliefs to find any association between vaccination uptake during the last five years and future vaccination acceptance	Multicenter cross-sectional study	Health care students
Ruggeri, M., 2020 [13]	European perspective (Italy)	To test an analytical framework developed for the estimation of the fiscal impacts of vaccination programs for influenza, pneumococcus, and HZ in Italy	Economic evaluation (fiscal impact model)	Workers (15–64 years old)
Grieco, L., 2020 [44]	European perspective (UK)	To identify the range of pandemic and policy scenarios under which plans to immunize the general UK population exist	Epidemiological modeling and health economic analysis	General population
Scholz, S.M., 2021 [45]	European perspective (Germany)	To examine the cost-effectiveness of a possible extension of the recommendation to include strategies of childhood vaccination against seasonal influenza using QIVs	Economic evaluation (dynamic transmission model, from a societal and a third-party payer perspective)	Pediatric population
Annemans, L., 2021 [46]	European perspective (Belgium)	To highlight which particular value elements of vaccination remain neglected in economic evaluations	Expert opinion	General population
Antonelli-Incalzi, R., 2021 [47]	European perspective (Italy)	To provide an overview of the existing evidence on the value of adult vaccination in the Italian context	Literature overview	Elderly individuals
Macias, A.E., 2021 [48]	Global perspective	To outline the main influenza complications and societal impacts beyond the classical respiratory symptoms of the disease	Literature review	General population
Aznab, M., 2021 [49]	Non-European perspective (Iran)	To evaluate the value of influenza vaccination in the cancer population	Cross-sectional descriptive study	Cancer population
Villani, L., 2022 [3]	European perspective (Italy)	To summarize the literature regarding the costs of pediatric influenza in Europe, paying particular attention to the direct and indirect costs considered in the economic evaluations	Systematic review	Pediatric population
Calabrò, G.E., 2022 [4]	European perspective (Italy)	To carry out an HTA of the aQIV	HTA	Elderly individuals

QIV: quadrivalent influenza vaccine; TIV: trivalent influenza vaccine; DALY: disability-adjusted life year; RV: rotavirus; MD: meningococcal disease; PD: pneumococcal disease; HPV: human papillomavirus; Hep B: hepatitis B; HZ: herpes zoster; HCWs: health care workers; LICs: low income countries; ILI: influenza-like illness; VPDs: vaccine-preventable diseases; IVE: influenza vaccine effectiveness; HTA: health technology assessment; aQIV: adjuvanted quadrivalent influenza vaccine.

**Table 2 vaccines-10-01675-t002:** Main findings on influenza vaccination values of the included studies.

First Author, Year [Ref]	European Perspective, Non-European Perspective or Global Perspective (Country)	Personal Value (Clinical Outcomes, Patient-Reported Outcomes, Patient-Reported Experience Measures, Citizens’ Involvement and Empowerment in Vaccination)	Allocative Value (Accessibility, Equity, Affordability, Appropriateness, Unwarranted Variations, Innovation)	Technical Value Type of Study/Economic Model, Drivers of Costs, Innovative Vaccination/Vaccine-Related Cost Drivers)	Societal Value (Population’s Wellbeing, Indirect/Community Protection, Shared Decision-Making Process)	Other Aspects of Influenza Vaccination Value	Main Reflections/Actions
Bailey, T.M., 2011 [18]	Non-European perspective (Canada)		X (Equity)		X (Shared decision-making process)		It is crucial to formulate fairness-based immunization plans. Conducting prepandemic research is essential to engage the public, educate them, and solicit citizen feedback. Immunization plans based on shared values and a fair process are those that will be most successful during an emergency.
Lee, B.L., 2011 [19]	Non-European perspective (USA)			X (Type of study/economic model)			A universal vaccine could provide substantial economic value by exceeding current annual vaccine limits. Investments in the development of universal vaccines must be encouraged.
Luyten, J., 2011 [20]	Global perspective	X (Clinical outcomes)	X (Accessibility, equity)		X (Population’s wellbeing, indirect/community protection)		Vaccination policy is an ethically challenging area of public policy. It is a question of relevant importance for the community that goes beyond individual-based ethics.
Lee, B.Y., 2012 [21]	Non-European perspective (USA)			X (Type of study/economic model)			The addition of the influenza B strain to convert the TIV into a QIV could result in substantial cost savings to society and third-party payers, even when the cost of QIV is substantially higher.
de Waure, C., 2012 [22]	Global perspective			X (Type of study/economic model)			Influenza vaccination among elderly and high-risk groups is a cost-effective intervention; however, a standardization of methods is necessary to ensure comparability and transferability of the economic model results.
Mamma, M., 2013 [23]	European perspective (Greece)			X (Type of study/economic model)			Providing a vaccination program against seasonal and pandemic A/H1N1 influenza can incur a substantial net benefit for customs officers. However, the size of the benefit strongly depends upon the attack rate of influenza, the symptomatic rate, and the participation rate of the customs officers in the program.
Nagata, J.M., 2013 [24]	Global perspective					X (Societal and cultural values and health beliefs about influenza vaccination)	Incorporating a framework that takes into account societal determinants of health in vaccine policies may foster immunization equity among the most vulnerable populations.
Preaud, E., 2014 [8]	European perspective (27 European countries: Austria, Belgium, Bulgaria, Cyprus, Czech Republic, Denmark, Estonia, France, Finland, Germany, Greece, Hungary, Ireland, Italy, Latvia, Lithuania, Luxembourg, Malta, the Netherlands, Poland, Portugal, Romania, Slovakia, Slovenia, Spain, Sweden, UK)	X (Clinical outcomes)		X (Type of study/economic model)	X (Population’s wellbeing)		- Development of more effective vaccines for elderly individuals is needed. - Improvements in vaccination coverage among nonelderly individuals and improvements in vaccine effectiveness among elderly individuals are needed to improve vaccination program effectiveness. - Public health and economic benefits from seasonal influenza vaccination can be substantially increased if a 75% vaccination coverage rate is reached: twice as many cases could be prevented and hundreds of thousands of hospitalizations and physician visits could be avoided, reducing the burden on health care systems. - Immunization programs for specific target populations are needed, as well as the implementation of evidence-based vaccination policies. In parallel, the medical community and the vaccine industry should continue to invest in R&D to produce new flu vaccines. - Full implementation of current influenza vaccination recommendations could reduce the influenza burden and increase appropriate health care resource allocation and support economic growth by preventing loss of productivity and preserving health.
Rappuoli, R., 2014 [25]	Global perspective					X (Importance of the full value of vaccination)	It is necessary to develop new technologies and health economic models capable of capturing not only the cost-benefit of vaccination but also the value of health itself.
Kristensen, M., 2016 [26]	European perspective (the Netherlands)					X (Vaccination value for older adults)	Knowing the influenza burden allows us to consider the added value of vaccination among elderly individuals and will help in defining priorities in immunization programs.
Uhart, M., 2016 [27]	European perspective (France, Germany, Italy, Spain and UK)			X (Type of study/economic model)			It is estimated that, compared to TIV, QIV may result in a substantial decrease in epidemiological burden and flu-related costs.
Barbieri, M., 2016 [28]	European perspective (Belgium, France, Germany, Italy, Spain, Sweden, the Netherlands, UK)			X (Type of study/economic model)			Vaccination is cost-effective in all included studies (the only exception is a UK study) with better results from the societal perspective.
Gibson, E., 2016 [29]	European and non-European perspectives (Europe and USA)			X (Type of study/economic model)			- The economic impact of a pediatric influenza immunization program was influenced by vaccine efficacy, immunization coverage, costs, and most significantly by herd immunity. - Pediatric influenza immunization may offer a cost-effective strategy when compared with HPV and varicella immunization and possibly more value compared with other childhood vaccines (RV, Hep B, MD, and PD).
McElhaney, J.E., 2016 [30]	European perspective (France, Belgium)	X (Clinical outcomes)			X (Population’s wellbeing, indirect/community protection)		- Protection may be improved by offering vaccination to younger, healthier individuals before they become frail. In addition, offering vaccination to caregivers and others who are in contact with elderly individuals could also improve protection. - Health care providers and public health decision-makers need to understand the vaccination value more in depth and see it as an important preventive tool in promoting successful aging.
Poscia, A., 2016 [31]	European perspective (Italy)			X (Drivers of costs)	X (Population’s wellbeing, indirect/community protection)		Emphasizing the societal and economic values of flu vaccination is needed. Communication and the awareness of VPDs, such as influenza, in the general community is an important starting point. Health care professionals and public health/social workers can play a key role in this regard.
Wilder-Smith, A., 2017 [5]	Global perspective	X (Clinical outcomes)	X (Equity)	X (Type of study/economic model)	X (Population’s wellbeing, indirect/community protection)		- Dynamic models should be prioritized over static models, as the constant force of infection assumed in static models will usually underestimate the effectiveness and cost-effectiveness of the immunization program by underestimating indirect effects. - The economic impact of vaccination should incorporate health and no health benefits of vaccination for both the vaccinated and unvaccinated populations, thus allowing for estimation of the societal value of vaccination. - The full benefits of vaccination go beyond the direct prevention of etiologically confirmed diseases and often extend throughout a vaccinated person’s lifetime, prevent community outcomes, support the sustainability of health systems, promote health equity, and benefit local and national economies.
Lorenc, T., 2017 [10]	Global perspective				X (Indirect/community protection)		HCWs may be motivated to accept vaccination to protect themselves and their patients against infection. However, several beliefs may be barriers to vaccine uptake, including concerns about side effects, skepticism about vaccine effectiveness, and the belief that influenza is not a serious illness.
Ting, E.E.K., 2017 [12]	Global perspective			X (Type of study/economic model)	X (Population’s wellbeing, indirect/community protection)		- From the societal perspective, vaccination is cost-effective for children, pregnant and postpartum women, high-risk groups, and in some cases, healthy working-age adults. - Immunization programs using group administration are more cost-effective than programs using individual administration. - The perspective, programmatic design, setting, and inclusion of herd immunity affects cost-effectiveness.
Ortiz, J.R., 2017 [32]	Non-European perspective (LICs)	X (Clinical outcomes)	X (Accessibility, equity)		X (Indirect protection)		- Maternal flu immunization prevents influenza in two high-risk groups (mothers and their infants). - Strengthening maternal immunization in LICs could improve health systems for antenatal care delivery, build a platform for other vaccines to be used during pregnancy, and strengthen systems to regulate, procure, and distribute flu vaccines in response to a future pandemic.
Wagner, M., 2017 [33]	European perspective (England)					X (Evaluation of innovative methods of influenza surveillance through social media)	Implementation of a program for school-age children can be supported and evidence of the vaccination value can be provided using social media as an additional flu surveillance tool.
Esposito, S., 2018 [11]	European perspective (Italy, France, Spain, Germany, UK)					X (Value of influenza vaccination for elderly individuals)	Influenza in elderly individuals represents a substantial health and societal burden. Vaccination is a value intervention to prevent influenza in Europe.
Meijboom, M.J., 2018 [34]	European perspective (the Netherlands)			X (Type of study/economic model)	X (Indirect/community protection)		In addition to the decreased burden of patient morbidity among hospitalized patients, the effects of a hospital immunization program slightly outweigh the economic investments. These outcomes may support health care policy-makers’ recommendations about the influenza vaccination program for HCWs.
Boey, L., 2018 [35]	European perspective (Belgium)	X (Citizen, in particular HCWs, involvement and empowerment in vaccination)			X (Indirect/community protection)		- Seasonal influenza campaigns are needed, in which education, communication, and easily accessible vaccination are promoted. - It is important to focus not only on the value for patients during flu vaccination campaigns but also on the personal benefits for the HCWs themselves.
Li, K.K., 2019 [36]	Non-European perspective (China)				X (Indirect/community protection)	X (Cultural values, perceived social benefits, and social influence regarding influenza vaccination among nurses)	Collectivism may guide how nurses attend to and process social information and subsequently influence their vaccination adoption behaviors.
Hollingsworth, R., 2020 [37]	Global perspective					X (Need for an appropriate methodology to assess the full value of flu vaccination and support evidence-based decision-making)	- Better interpretation of IVE will improve the broader assessment of the value of influenza vaccination. - A hybrid experimental-observational method, referred to as a “pragmatic clinical trial” (PCT), has recently been used to estimate IVE. This method prospectively investigates randomized groups but measures endpoints from routinely collected data or vital statistics. The primary advantage of PCTs is that they can be designed to be more reflective of real-world vaccine experiences, with research questions (such as outcomes, patient populations, and so on) that are more relevant to policy-makers and clinical decision-makers.
Domnich, A., 2020 [38]	European perspective (Italy)					X (Value of flu vaccines from the citizens’ perspectives)	To increase influenza vaccination coverage rates, multidisciplinary-targeted interventions are needed. The role of general practitioners is crucial in increasing influenza vaccine awareness and acceptance by effective counseling.
Calabrò, G.E., 2020 [39]	European perspective (Italy)					X (HTA is an evidence-based tool to assess the value of influenza vaccines)	HTA is an evidence-based tool for assessing the value of vaccines and supporting decision-making in vaccination strategies. However, the success of flu vaccination also depends on the empowerment and involvement of citizens in the decision-making process.
Ruscio, B., 2020 [40]	Global perspective					X (Value of a coordinated work plan for epidemic and pandemic influenza vaccine preparedness)	- It is essential to promote the strengthening of seasonal influenza immunization programs in LICs and LMICs to reduce the clinical and economic burden of annual influenza outbreaks and to mitigate the threat of future pandemics and improve global health safety. - For adequate preparation for influenza pandemics, the following actions are necessary: - greater collaboration between public (government) and private sectors; - adequate communication on the health risks associated with influenza and on the efficacy and value of vaccines; and - greater collaboration for the definition of a pandemic plan shared between different international stakeholders (government and nongovernment organization representatives, civil society representatives, vaccine manufacturers, international organizations, and health security and influenza experts).
Aidoud, A., 2020 [41]	European perspective (France)					X (Influenza vaccination value for older adults at high risk of flu infection and CHD complications)	The greatest benefit of flu vaccination is the prevention of infection and its cardiovascular complications in elderly individuals. Research on the molecular immunology of the response to flu vaccination and its correlation with atheroprotective processes should be further implemented.
Valentino, S., 2020 [42]	Extra-European perspective (USA)					X (Value of educational interventions for nursing students)	Specific educational interventions aimed at nursing students can lead to an improvement in knowledge about influenza and to an increase in vaccination coverage in this target population.
Cella, P., 2020 [43]	European perspective (Italy)					X (Vaccination value for health care students)	Health care students must be considered a priority group to be actively involved in campaigns promoting vaccination.
Ruggeri, M., 2020 [13]	European perspective (Italy)			X (Type of study/economic model, drivers of costs, innovative vaccination/vaccine-related cost drivers)			Decision-makers should adopt a full societal perspective to assess the economic value of vaccines. Traditional methods to estimate the cost of illness from a social perspective can be improved by additionally considering the fiscal impact, which accounts for the decrease in fiscal revenues due to disease. The potential reduction of the fiscal impact should be included in the assessment of new health technologies, adding a new dimension to this valorization.
Grieco, L., 2020 [44]	European perspective (UK)			X (Type of study/economic model)			Plans based on the responsive purchase of vaccines have wider benefits than plans reliant on the purchase and maintenance of a stockpile if immunization can start without extensive delays. This finding depends on whether the responsively purchased vaccines are driven by avoiding the costs of storing and replenishing a stockpile.
Scholz, S.M., 2021 [45]	European perspective (Germany)			X (Type of study/economic model)	X (Indirect/community protection)		The introduction of any routine childhood vaccination strategy with QIV will be cost saving from a societal perspective. Vaccinating the age group 2 to 17 years of age seems to offer the highest health benefits.
Annemans, L., 2021 [46]	European perspective (Belgium)	X (Clinical outcomes)	X (Equity)	X (Type of study/economic model, drivers of costs, innovative vaccination/vaccine-related cost drivers)	X (Population’s wellbeing, indirect/community protection,		- Decision- and policy-makers must be made aware of the limitations of traditional economic models to assess the value of vaccines. - Future economic evaluations need to include the effect of vaccination on preventing complications, on generating health gains for caregivers, and on community benefits beyond individual protection. Furthermore, - guidelines are needed for the economic evaluation of the full value of vaccinations; - economic evaluations need to be conducted from the societal perspective, rather than the health care payer perspective, in order to assess the full value of vaccines; and - economic evaluations need to consider how costs and health outcomes associated with vaccination contribute to broader goals of social equity and ensure optimal population health.
Antonelli-Incalzi, R., 2021 [47]	European perspective (Italy)					X (Value of adult immunization)	- All stakeholders involved in the vaccination process (Health authorities and health institutions, HCWs, the public, and industries) should work together to ensure that people have healthy lives. - The benefits of vaccination such as influenza vaccination should become a major topic of conversation beyond the current pandemic context. - A paradigm shift is necessary, in which citizens are informed by HCWs about relevant vaccines before they become sick.
Macias, A.E., 2021 [48]	Global perspective			X (Type of study/economic model)	X (Population’s wellbeing, indirect/community protection)		Recognizing the broader consequences of influenza infection is essential to determine the full burden of disease in different subpopulations and the value of prevention.
Aznab, M., 2021 [49]	Extra-European perspective (Iran)	X (Clinical outcomes)				X (Influenza vaccination value in cancer population)	Cancer patients are recommended to receive the flu vaccine usually during the flu epidemic season to reduce mortality.
Villani, L., 2022 [3]	European perspective (Italy)	X (Clinical outcomes)		X (Type of study/economic model, drivers of costs)	X (Population’s wellbeing, indirect/community protection)		Knowing the pediatric influenza costs could be useful for decision-makers to ensure better resource allocation for prevention and implement value-based immunization strategies.
Calabrò, G.E., 2022 [4]	European perspective (Italy)	X (Clinical outcomes, Citizen involvement and empowerment in vaccination)	X (Appropriateness, equity)	X (Type of study/economic model, drivers of costs)	X (Population’s wellbeing, indirect/community protection)		- The use of aQIV in the elderly population is cost-effective and moves in the direction of strengthening the appropriateness of the use of available influenza vaccines. - The HTA approach could help to appraise the broad value of vaccines.

QIV: quadrivalent influenza vaccine; TIV: trivalent influenza vaccine; R&D: research and development; RV: rotavirus; MD: meningococcal disease; PD: pneumococcal disease; HPV: human papillomavirus; Hep B: hepatitis B; HZ: herpes zoster; VPDs: vaccine-preventable diseases; HCWs: health care workers; LICs: low income countries; IVE: influenza vaccine effectiveness; PCT: pragmatic clinical trial; LMICs: low-middle income countries; CHD: coronary heart disease.

## Data Availability

Not applicable.

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
