# Peer review of "Influenza Vaccination Assessment according to a Value-Based Health Care Approach"

_vaccines, 2022, doi:10.3390/vaccines10101675_

Round 1
Reviewer 1 Report
The authors, in a Review Paper, discuss on the impacts of influenza on public health and the obtained results due to vaccination based on different research studies (40 studies). They consider a new value structure of four pillars such as personal, allocative, technical and societal value. By querying three databases, they conduct a systematic review and then find considerable results on the influenza vaccination.
The article gives a complete background and history of different studies in this regard. Materials and methods are described well. The main characteristics of different studies can be found in some tables explaining all details. As a review paper, it has the required standards and describe all results and discussions completely.
English of the paper is good, but it is better that the authors check it once again carefully. Also, please check all details of references. The topic is in the scope of the journal.
In general, I recommend it for publication in the journal “Vaccines”.
Author Response
We thank the Reviewer for his/her comments and suggestions. We checked the English and references as required and made some changes. We also specify that the text has been subjected to linguistic revision as specified in the acknowledgments.

Reviewer 2 Report
This article focused on the whole value of influenza vaccination and can promote the correct use of flu vaccines, and it' s very valuable.
I think this study can be greatly strengthened by addressing completely recognition the whole value of influenza vaccination can promote vaccine acceptance and vaccination coverage.
Author Response
We thank the Reviewer for his/her comments and suggestions. We included a sentence in the conclusions to emphasize what the reviewer suggested.

Reviewer 3 Report
This paper presents a comprehensive viewpoint on vaccine effectiveness and societal value.
In several instances the authors mention the importance of assessing the full societal value and the related economic impact. While I think their viewpoint has some validity, one should not that vaccination campaigns launched by public health campaigns ARE focused on community/societal value, not just the individual. Therefore, to imply that is hardly exists is not entirely true.
The authors failed to suggest what actions, or aspects of marketing could be used to increase vaccination among the underrepresented groups.
I agree that there is a dearth of publications on the societal value of vaccinations, however, I do not think it is because there is a deficit. It is quite possible that because this is so complex to capture, not much vaccine effectiveness work encompasses this. I think this is something the authors should consider and note as a possible limitation in the field of research.
Overall, I think this paper gives a lot for epidemiologists and other public serving officials to put to practice.
I have no specific suggested edits for any of the text within this article.
Author Response
We thank the Reviewer for his/her comments. We integrated the discussion of our manuscript including as a limit the difficulty of evaluating the whole value of the flu vaccination. In particular we wrote "Moreover, it is important to underline that the estimation of the whole value of the influenza vaccination is particularly complex and, therefore the scarcity of evidence recovered in the literature, especially for some value pillars, could be linked to this difficulty of evaluation. Our paper describes the current evidence on the assessment of personal, allocative, technical and societal value of influenza vaccination and stresses the need to expand these aspects as well in the research field. As presented, the assessment of the whole value of flu vaccination needs to consider not just the direct impact on health and health care but also the wider impact on economic growth and societies. These wider impacts, although difficult to measure and still under investigated, should be taken into consideration to better depict the whole value of flu vaccines and vaccination and counteract vaccines hesitancy and misuse”.
